# Physical Functions among Children before and during the COVID-19 Pandemic: A Prospective Longitudinal Observational Study (Stage 1)

**DOI:** 10.3390/ijerph191811513

**Published:** 2022-09-13

**Authors:** Tadashi Ito, Hideshi Sugiura, Yuji Ito, Sho Narahara, Koji Noritake, Daiki Takahashi, Kentaro Natsume, Nobuhiko Ochi

**Affiliations:** 1Three-Dimensional Motion Analysis Laboratory, Aichi Prefectural Mikawa Aoitori Medical and Rehabilitation Center for Developmental Disabilities, Okazaki 444-0002, Japan; 2Department of Integrated Health Sciences, Graduate School of Medicine, Nagoya University, Nagoya 461-8673, Japan; 3Department of Pediatrics, Graduate School of Medicine, Nagoya University, Nagoya 466-8550, Japan; 4Department of Pediatrics, Aichi Prefectural Mikawa Aoitori Medical and Rehabilitation Center for Developmental Disabilities, Okazaki 444-0002, Japan; 5Department of Orthopedic Surgery, Aichi Prefectural Mikawa Aoitori Medical and Rehabilitation Center for Developmental Disabilities, Okazaki 444-0002, Japan

**Keywords:** COVID-19, children, balance functions, obesity, life habits, physical functions, pandemic

## Abstract

One major impact of the COVID-19 pandemic on children’s lifestyles is the lack of exercise owing to activity restrictions. However, information regarding the way in which physical functions among children decline under these circumstances remains scarce. In this study, we compared the physical functions and life habits among children before and during the COVID-19 pandemic in Japan. The participants involved 40 children aged between 9–15 years (23 girls and 17 boys) who were examined medically both before and during the pandemic. The compared variables included muscle strength, static and dynamic balance functions, gait speed, body fat percentage, screen and sleep times, quality of life, and physical activity time. During the pandemic, compared to before the pandemic, children had lower levels of dynamic balance functions (*p* = 0.039), increased body fat percentages (*p* < 0.0001), longer screen time per day (*p* = 0.002), and shorter sleep time per day (*p* < 0.0001). Between the two periods, there were no significant differences in muscle strength, static balance functions, gait speed, quality of life, and physical activity time. The activity restrictions imposed as a result of the COVID-19 pandemic negatively affect dynamic balance functions, body-fat levels, and life habits among children.

## 1. Introduction

From December 2019, the coronavirus disease (COVID-19) has been affecting the health of individuals in various regions worldwide. In March 2020, the World Health Organization (WHO) declared COVID-19 a pandemic. Although Japan is currently in a quasi-state of emergency, since April 2020, other states of emergency have been repeatedly proclaimed by the government of Japan. People have been requested to practice social distancing and avoid non-essential social gatherings as well as non-urgent meetings. Since the onset of the COVID-19 pandemic in Japan, children have not been allowed to engage in team-based sports or outdoor activities, and as a result, their opportunities for engaging in various physical activities or physical education in their school environments remain limited [1,2]. As a result of the COVID-19 pandemic, prolonged restrictions imposed on the engagement in physical activities among children may significantly affect various aspects related to child development, including physical functions and life habits [3]. Additionally, physically inactive children may experience deteriorated physical functions or health problems later in life [3,4,5]. Therefore, the activity restrictions imposed as a result of the COVID-19 pandemic may exacerbate the risks for increased body fat percentages among children as well as the risk factors for childhood obesity [6]. Among children, the daily indulgence in digital media results in increased screen time, which could also result in poor quality of sleep among such individuals [7]. According to Wunsch et al., in the context of COVID-19-related restrictions on social activities among children, healthcare policy makers should implement approaches for reducing screen time among children [8]. Regarding dietary changes among children, an increase in the amount of food consumed at home has been reported during the COVID-19 pandemic [9,10]. Moreover, according to a study conducted in Germany, the quality of life among children and adolescents has declined during the COVID-19 pandemic [11].

Physical functions refer to the abilities allowing for the performance of physical activities among individuals. Such functions are evaluated using the single leg stance or one-legged stance test as well as the two-step test as measures of balance control, the five times sit-to-stand test (FTSST) as a measure of functional performance, and gait speed as a measure of gait ability [12,13]. Previous studies have demonstrated that the COVID-19 pandemic has made people highly prone to poor physical functions, irregular lifestyles, and obesity [3,14,15,16,17]. Children who continue to experience behavioral changes as a result of COVID-19-related restrictions on social-based physical activities may experience health problems later in life. Therefore, it is imperative to identify the signs of decreased physical functions and potentially harmful life habits among children before and during the COVID-19 pandemic. However, confirmations on the changes in physical functions and life habits among children remain inconsistent in the studies on this subject, as it pertains to the COVID-19 pandemic.

Therefore, further insight on the COVID-19 pandemic’s current impact on physical functions and life habits among children is crucial in ensuring the effective implementation of public health policies aimed at ensuring the provision of care to children after COVID-19. Few studies have investigated the differences in physical functions among children in Japan both before and during the COVID-19 pandemic by applying direct evaluation approaches. Since January 2018, for screening purposes, various researchers have been performing systematic physical evaluations among children to determine the aspects affecting their changes in physical functions [3]. Although the data obtained during the COVID-19 pandemic were not initially collected for investigation purposes, the access to such data provided an exceptional opportunity for conducting further studies on the effects of COVID-19 on the physical functions and life habits among children in Japan [3].

In this study, we investigated the effects of the COVID-19 pandemic on the physical functions and life habits among children in Japan. We examined the ways in which the physical functions and life habits among such children changed during the COVID-19 pandemic by comparing data obtained through medical examinations and physical-function evaluations conducted between two periods: before and during the COVID-19 pandemic.

## 2. Materials and Methods

### 2.1. Study Design and Population

We obtained medical-examination data from 60 participants before and during the COVID-19 pandemic in Okazaki city, Japan. Our objective was to investigate the influence of the COVID-19 pandemic on physical functions and life habits among children aged 9–15 years. This study was conducted in accordance with the guidelines provided in the Strengthening the Reporting of Observational Studies in Epidemiology (STROBE) statement. The data were obtained based on the frameworks used in other studies on the evaluations of physical functions and medical examinations conducted in Okazaki city, Japan [3,12]. In this prospective longitudinal observational study, the data obtained from the representative sample were collected periodically. The baseline data were collected between January 2018 and March 2020. During the COVID-19 pandemic, the data were collected between June 2020 and June 2022. The participants involved in this study were then requested to participate in our previous study on physical-function evaluation and medical examination to help assess the changes directly affecting their physical functions and lifestyle habits.

This study was conducted in accordance with the Declaration of Helsinki. For the evaluation of the physical functions, lifestyle habits, quality of life questionnaires, and medical examinations, approval was obtained from the Ethics Review Board of the corresponding author’s institution (no. 29002; approval date: October 10, 2017). Written informed consent and assent were obtained from all the children involved in this study as well as from their parents and/or guardians.

The evaluations comprised medical examinations conducted by pediatric neurologists and pediatric orthopedic surgeons, who read the questionnaires to the children involved in this study. One physical therapist and one research assistant evaluated the physical functions and life habits among the participants involved in this study. In this prospective longitudinal observational study, the baseline for physical functions, life habits, and the medical examinations conducted afterwards were evaluated by focusing on the physical functions and life habits of the children involved in this study. The physical function assessment and life habits questionnaires comprised 10 items obtained from the FTSST, the two-step test, the single leg stance test, gait analysis (gait speed), body fat percentage, physical activity time, screen time, sleep time, seven-day diet history involving three meals per day, and the Pediatric Quality of Life Inventory assessment tool (version 4.0).

From the 437 children eligible for participation in this study, only 60 who had participated in at least one medical examination before the COVID-19 pandemic were included in the sample. The data obtained from the participants were evaluated only once before and during the COVID-19 pandemic. The exclusion criteria included individuals suffering from orthopedic, neurological, respiratory, and ophthalmologic issues as well as those suffering from digestive system disorders and intellectual disabilities, as indicated through substandard scores on the Peabody Picture Vocabulary Test-Revised and Raven’s Colored Progressive Matrices [18,19], which could affect the results of physical function assessments, thereby generating incomplete data. As a result, 20 out of 60 children were excluded, thereby providing 40 participants who constituted the final sample used in this study (Figure 1).

### 2.2. Data Collection

#### 2.2.1. FTSST

The FTSST was used to evaluate functional muscle strength in the lower extremities [12]. Children had their arms crossed over their chest and were invited to stand up and sit down five times from the chair, as rapidly as possible without using arm supports [12,20]. The results of the FTSST were normalized using leg length, as expressed in the following formula: FTSST (s) = FTSST/SQRT (leg length/*g*), where *g* represents gravitational acceleration (9.81 ms^2^) [21].

#### 2.2.2. Two-Step Test

To examine the dynamic balance functions among the children involved in this study, the children were instructed to stand with their toes behind a starting line, take two steps, each as large as possible, and then align both of their feet [13]. The two-step test was evaluated using the ratio of distance (the length of the two steps [in centimeters]) to the height of the children (in centimeters) [13]. We conducted the measurements twice, and the maximum value was used for subsequent analysis.

#### 2.2.3. One-Legged Stance Test

The single leg stance test, which is also known as the one-legged stance test, was used to assessed for how long a participant could remain standing on a single leg with their eyes open. This test was used to measure the static balance functions among the participants [3,12]. The results of the tests are evaluated as the maximum number of seconds each participant spends standing on either their left or right leg. The assessment concluded when the lifted leg touched the supporting leg, the toes, when the other leg touched the floor, or following 120 s of maintaining balance successfully [3,12]. The results of the single leg stance test were normalized using leg length, as expressed in the following equation: single-legged standing time (s) = single-legged standing time/SQRT (leg length/g) [21].

#### 2.2.4. Gait Analysis

Gait analysis was conducted using the three-dimensional gait analysis system (sampling frequency: 100 Hz) involving eight cameras (MX-T 20S; Vicon, Oxford, UK) [3,12]. The Plug-In-Gait model (January 2018 and March 2020) and the Conventional Gait Model 2.3 (June 2020 and June 2022) were used to assess gait by patching a marker on the pelvis and lower extremities of the participants [3,12,22,23]. The participants were filmed as they walked barefoot at a usual gait speed [3,12,23]. Gait speed is a crucial component for evaluating physical functions, and it has been reported to be a basic indicator of gait development among children, which reflects the status of a healthy individual [24,25]. The mean of usual gait speed was calculated based on the results of the analyzed right and left lower extremities (three gait trials) [3,12,22]. The gait speed was normalized using leg length, as expressed in the following equation: gait speed (m/s) = speed/SQRT (leg length × *g*) [21].

#### 2.2.5. Body Fat Percentage

The multi-frequency bioelectrical impedance analyzer (MC-780; Tanita, Tokyo, Japan) oriented in an upright position was used to estimate the percentage of body fat among the participants [3,12]. The participants were instructed to stand with each of their hands outstretched to ensure that skin-to-skin contact was prevented. The participants were holding hand electrodes, and they maintained contact with the fore and posterior electrodes through the soles of their feet. The evaluation for body fat percentage was completed within 15 s. The resistance of the multi-frequency bioelectrical impedance analyzer was evaluated at 5, 50, and 250 kHz [3,12]. The multi-frequency bioelectrical impedance analyzer is non-invasive, and in previous studies, many researchers have used this device to ensure the convenient evaluation of body composition among children [26]. This evaluation was conducted at a minimum of two hours after meals, as recommended in the operating manual.

#### 2.2.6. Questionnaires

Among children, the WHO recommends a minimum of 60 min of moderate and vigorous physical activity per day for more than five days a week [27]. Moderately intensive physical activity time per week was assessed based on the physical activity index. The children participating in this study completed the Japanese version of Pediatric Quality of Life Inventory (version 4.0) on their own, as part of the quality-of-life component for evaluation [28]. To evaluate screen time, the participants were instructed to report their screen time (watching television and movies as well as using smartphones) per day, and their answers were confirmed by consulting their parents. Sleep time per day was assessed by parents and children in relation to the children’s sleep history [3,13]. Regarding the number of meals, the children and their parents were instructed to provide a seven-day dietary history involving three meals per day [3,29].

#### 2.2.7. Sample Size

G*Power (Heinrich Heine University of Düsseldorf, Düsseldorf, Germany) was used to ascertain the optimum sample size with effect size (d = 0.5), a two-tailed alpha value of 0.05, and a statistical power of 0.8 [30,31]. The effect size of 0.5 was determined based on the mean and standard deviations of the data associated with balance functions from previous studies conducted by Yanovich et al. during the COVID-19 pandemic [32]. Based on these assumptions, a desired sample size of 35 participants was included in the evaluations.

### 2.3. Statistical Analysis

Regarding statistical analysis, the data collected throughout this study were analyzed using SPSS Version 24.0 (IBM Corp., Armonk, NY, USA). The normal distribution of the variables was verified using the Shapiro-Wilk test. A chi-squared goodness-of-fit test was used to compare the differences in gender proportions. The paired-sample *t*-test and the Wilcoxon signed-rank test were used to determine whether statistically significant differences existed in the evaluated physical functions or life habits between the period before and during the COVID-19 pandemic. Two-sided *p*-values < 0.05 were considered statistically significant. Effect sizes with r = −0.1 or 0.1 were considered insignificant. Those with r = −0.3 or 0.3 were considered moderate, and those with r = −0.5 or 0.5 were considered highly significant.

## 3. Results

The participants’ demographic characteristics are listed in Table 1. Significant differences in height, weight, and body mass index were observed as they increased throughout the growth of the participants involved in this study. Throughout the sample population involved in this study, female children comprised 57.5% and male children comprised 42.5%. Additionally, throughout this study, the results of the chi-squared goodness-of-fit test did not demonstrate any significant differences related to gender (*p* = 0.343). Moreover, between the two periods: before and during the COVID-19 pandemic, there were no significant differences in the z-scores related to body mass indices among the participants.

Table 2 summarizes the results for the evaluation of physical functions and body composition among the participants involved in this study. The participants achieved lower scores in the two-step test (*p* = 0.039). During the COVID-19 pandemic, compared to before the pandemic, the participants involved in this study demonstrated increased levels of body fat percentages (*p* < 0.0001). Between the two periods (before and during the COVID-19 pandemic), there were no significant differences in the results of the FTSST, the single leg stance test, and gait speed.

The results of each questionnaire are summarized in Table 3. Compared to before the COVID-19 pandemic, the children involved in this study had longer screen time per day (*p* < 0.001) during the pandemic. Moreover, these individuals had shorter sleep times (*p* < 0.0001) before the COVID-19 pandemic, compared to their sleep times during the pandemic. Between the two periods (before and during the COVID-19 pandemic), there were no significant differences in the results on physical activity-related evaluations, those based on the Pediatric Quality of Life Inventory (version 4.0), or those based on the number of meals per day among the participants involved in this study.

## 4. Discussion

The main objective of this study involved analyzing the physical functions and life habits among children in Japan before and during the COVID-19 pandemic. The main results of this study showed that dynamic balance among the children involved in this study was affected during the pandemic, as indicated by the deteriorations presented through the results of the two-step test. A previous study conducted by Martínez-Córcoles et al. [33] found that limited time and opportunities for children to engage in social-based physical activities may result in decreased balance among them. In this study, we speculated that this result may be because of inadequate opportunities for children to engage in learning exercises aimed at ensuring the further development of their dynamic balance functions. Other studies have reported that the failure to engage in regular social-based physical activities in children or their decreased participation in such activities may result in decreased balance functions [34,35]. Meanwhile, according to a study conducted by Condon et al., the time spent standing on one leg increases as children become older, especially between seven and eight years of age [36]. However, the results of this study showed no improvement in the time spent standing on one leg with the increase in age among such individuals, thereby suggesting that the effects of the restrictions on social-based physical activity imposed during the COVID-19 pandemic may hinder the normal development of static balance functions among children.

Moreover, according to a study conducted by Rúa-Alonso et al., the social constraints resulting from the COVID-19 pandemic may negatively affect body composition (increased obesity and body mass index) and result in decreased muscle strength among children and adolescents [17]. However, currently, no consensus has been reached with regard to this aspect. Other studies reported measures related to the mitigation of COVID-19 infections, which seemed to have no long-term negative effects on children in terms of muscle strength [37]. Therefore, during the COVID-19 pandemic, it may be speculated that the levels of physical activity among children remain insufficient for ensuring the proper activation of their balance functions. This fact might result in movement disorders that consequently affect the health statuses of children. Therefore, it is necessary to prevent the lack of social-based physical activities among children and motivate such individuals to engage in team-based sports after COVID-19. Hence, looking at the poor physical functions among children as a result of the COVID-19-related restrictions on social interactions, the results of this study suggest the need for implementing specific programs aimed at enhancing balance in school environments after the COVID-19 pandemic.

The results of this study revealed no significant differences in physical activity time, thereby suggesting that the deterioration of physical functions among children may be related to the quality of exercise rather than time itself. Furthermore, a recent study reported that the locations for physical activities among children also changed drastically as a result of the COVID-19 pandemic, with children engaging in physical activities at home, on sidewalks, and roads [38]. Future studies on this subject should consider analyzing the intensity of and contents associated with physical activity among children during pandemics, such as COVID-19. Additionally, children with longer screen times, shorter sleep times, and high levels of body fat percentage during the pandemic present differences that are statistically significant. This could be because of the limited opportunities resulting from the decreased engagement in school-based club activities and sports, thereby shortening outdoor playtime, with the main aim being the prevention of the spread of infection by limiting exercise tasks, such as minimizing contact with others when it comes to engaging in physical activities which results in an irregular life pattern that emerged during the COVID-19 pandemic. Moreover, regarding the increase in body fat percentage among children, there was no significant difference in physical activity time and the number of meals, thereby suggesting that irregular lifestyles, such as increased screen time and the insufficient ability to ensure the high quality of exercise among children, may be the cause behind this result.

Gait speed did not demonstrate any changes between the periods before and during the COVID-19 pandemic. Previous studies involving children aged between six and 12 years of age reported no significant differences in gait speed, whether such individuals met the criteria of at least one hour of physical activity per day or not. The results indicated that children with limited opportunities for engaging in physical activities did not necessarily present significant changes in their gait speed [12].

The quality of life and the number of meals among children, which were assessed using questionnaires based on the Pediatric Quality of Life Inventory (version 4.0) and the number of meals, remain unaffected during the COVID-19 pandemic. This result was similar to those of previous studies comparing children aged between six and seven years before and after the emergency declarations [3]. Therefore, children in Japan may be less sensitive to changes due to the COVID-19 pandemic in terms of quality of life and the number of meals.

There are some limitations of this study that must be considered. First, this study was conducted with a gap of two to three years between the two points for evaluation: before and during the COVID-19 pandemic. Additionally, except in the locations used for the evaluation environments, some participants could not be evaluated under similar conditions, such as time of day and season. Therefore, there may be some residual confounding impacts of age, time of day, and season on the findings. Second, the levels of physical activity used in this study were assessed using questionnaires and not through accelerometry. Finally, because this study was not based on repeated measures, the effects of the COVID-19 pandemic on balance functions, muscle strength, and life habits among children were only partially clarified. Future studies should focus on increased sample sizes to determine whether training that is specific to balance functions among children is effective.

## 5. Conclusions

Generally, the study established that there was a significant decrease in dynamic balance functions, sleep time, and an overall increase in body fat and screen time among children. During the COVID-19 pandemic, children’s exercise tasks were often limited in order to prevent the spread of infection. Specifically, children had limited engagement in outdoor physical activities, school-based club activities, and sports. This led to inadequate opportunities for physical education, which significantly affected the physical functions among children in Japan. Furthermore, the decrease in balance functions among such individuals was not influenced by physical activity time during the COVID-19 pandemic.

## Figures and Tables

**Figure 1 ijerph-19-11513-f001:**
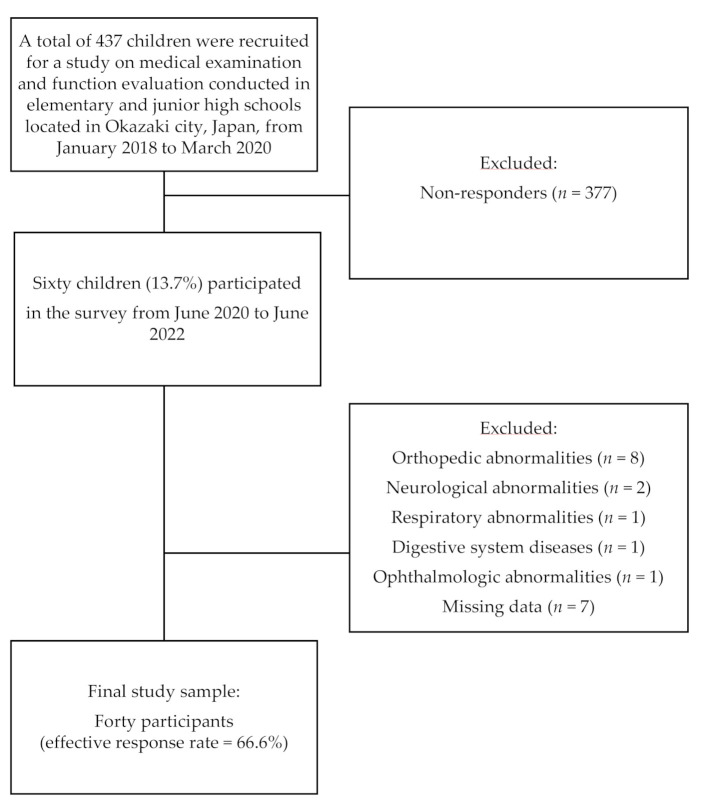
Flowchart of this prospective longitudinal observational study.

**Table 1 ijerph-19-11513-t001:** Participants’ demographic characteristics before and during the COVID-19 pandemic (*N* = 40; 23 girls and 17 boys).

Variable	Before the Pandemic	During the Pandemic	*p*	Effect Size (*r*)
Age (years), median (range)	9.0 (6–12)	12 (9–15)	0.0001	−0.9
Height (cm), mean (SD)	133.9 (9.7)	149.3 (8.8)	0.0001	1.0
Weight (kg), mean (SD)	28.8 (6.1)	39.3 (7.9)	0.0001	1.0
Body mass index z-score, mean (SD)	−0.6 (0.8)	−0.5 (0.8)	0.504	0.1

The *p*-value for height and weight were calculated using the paired-samples *t*-test, and the other p-values were calculated using the Wilcoxon signed-rank test.

**Table 2 ijerph-19-11513-t002:** Participants’ physical functions and body composition before and during the COVID-19 pandemic.

Variable	Before the Pandemic	During the Pandemic	*p*	Effect Size (*r*)
Five Times Sit-To-Stand test	21.1 (13.11–34.97)	20.1 (14.8–34.71)	0.064	−0.3
Two-step test	1.60 (0.14)	1.55 (0.1)	0.039	0.3
Single leg stance test	445.4 (39.4−502.3)	431.1 (136.1−473.5)	0.861	−0.03
Gait speed	0.46 (0.07)	0.45 (0.05)	0.415	0.1
Body fat percentage (%)	13.2 (4.5−32)	16.2 (4.0−37.6)	0.0001	−0.7

Data are presented as means (standard deviations), medians (ranges), and frequency (%). The *p*-values of the gait speed and the two-step test were calculated using the paired-samples *t*-test, and the other *p*-values were calculated using the Wilcoxon signed-rank test.

**Table 3 ijerph-19-11513-t003:** Participants’ questionnaire scores before and during the COVID-19 pandemic.

Variable	Before the Pandemic	During the Pandemic	*p*	Effect Size (*r*)
Physical activity (hour), median (range)	3.0 (0–20)	5.5 (0–20.5)	0.068	−0.3
Pediatric Quality of Life Inventory (version 4.0) (points), median (range)	95.7 (48.9–100)	95.7 (58.7–100)	0.371	−0.1
Number of meals (time), median (range)	21 (21–21)	21 (19−21)	0.102	−0.3
Screen time (hour), median (range)	1.5 (0.5–7)	2.0 (0.3–8.0)	0.002	−0.5
Sleep time per day (hour), median (range)	9 (8−10)	8 (6−9)	0.0001	−0.7

The *p*-values were calculated using the Wilcoxon signed-rank test.

## Data Availability

All relevant data are presented in the manuscript. All data are available from the authors upon reasonable request.

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
