# Peer review of "Physical Functions among Children before and during the COVID-19 Pandemic: A Prospective Longitudinal Observational Study (Stage 1)"

_ijerph, 2022, doi:10.3390/ijerph191811513_

Round 1
Reviewer 1 Report
Physical functions of children prior to and throughout the COVID-19 pandemic: a cohort study (Stage 1)
This paper reports the results of childrens’ physical function and life habits before and during the Covid-19 pandemic.
Introduction
The introduction is too brief and could be improved by addressing why physical function and life habits were chosen as the variables of investigation. Also, need to explain what are the components of physical function.
Secondly, the authors should mention and introduce studies that already recorded a decline in childrens’ physical levels, and explain this problem in more detail.
Finally, the whole introduction mentions 2 articles in related to screen time and only 1 article related to sleep but there is nothing discussed about other life habits (quality of life, meals etc.).
Methods
Perhaps divide the study design and population to different headings.
The second data point was collected in a 2year period. Were conditions the same throughout the 2 years, or was it different? (Other than citing age as a limitation)
How did the authors decide on the effect size values to determine sample size? (and later cite that as a limitation?)
Explain how gait speed is related to physical function.
Results
Check footnote under Table 3.
Discussion
In Line 210 the results of dynamic balance was presented, but the next sentence is about standing on one leg, The sequencing of sentences is confusing. Perhaps follow results with the supporting articles, followed by the discussion.
Line 216, how did the data show limited number of places to engage in physical activity?
“Limited opportunities” was mentioned several times as the reason for declining results. Describe limited opportunities in detail.
The results of Quality of life and meals are presented, but there are no discussions about the reasons there were no changes before and during Covid.
No proper discussion on body fat percentage.
There needs to be a stronger justification for including gait analysis in this study (only for gait speed?)
Author Response
Point-by-Point Responses
Reviewer #1
Thank you for providing us with insightful comments regarding our manuscript. We express our sincere gratitude for the time and effort required to review our study. Your comments and suggestions have helped immensely in improving our work.
Comment_1
Physical functions of children prior to and throughout the COVID-19 pandemic: a cohort study (Stage 1)
This paper reports the results of childrens’ physical function and life habits before and during the Covid-19 pandemic.
Response: Thank you for providing this accurate summary of our work. Each of your concerns has been addressed in greater detail below.
Comment_2
The introduction is too brief and could be improved by addressing why physical function and life habits were chosen as the variables of investigation. Also, need to explain what are the components of physical function.
Response: Thank you for pointing this out. We agree with the reviewer’s suggestion. This has now been added to the introduction.
“Regarding dietary changes among children, an increase in the amount of food consumed at home has been reported during the COVID-19 pandemic [9,10]. Moreover, according to a study conducted in Germany, the quality of life among children and adolescents has declined during the COVID-19 pandemic [11].
Physical functions refer to the abilities allowing for the performance of physical activities among individuals. Such functions are evaluated using the single leg stance or one-legged stance test as well as the two-step test as measures of balance control, the five times sit-to-stand test (FTSST) as a measure of functional performance, and gait speed as a measure of gait ability [12]. Previous studies have demonstrated that the COVID-19 pandemic has made people highly prone to poor physical functions, irregular lifestyles, and obesity [3, 13–16]. Children who continue to experience behavioral changes as a result of COVID-19-related restrictions on social-based physical activities may experience health problems later in life. Therefore, it is imperative to identify the signs of decreased physical functions and potentially harmful life habits among children before and during the COVID-19 pandemic. However, confirmations on the changes in physical functions and life habits among children remain inconsistent in the studies on this subject, as it pertains to the COVID-19 pandemic.”
Comment_3
Secondly, the authors should mention and introduce studies that already recorded a decline in childrens’ physical levels, and explain this problem in more detail.
Response: We agree with the reviewer’s comment. This discussion has been corrected in accordance with the reviewer’s comment.
“Previous studies have demonstrated that the COVID-19 pandemic has made people highly prone to poor physical functions, irregular lifestyles, and obesity [3, 13–16]. Children that continue to experience behavioral changes as a result of COVID-19-related restrictions on social-based physical activities may experience health problems later in life. Therefore, it is imperative to identify the signs of decreased physical functions and potentially harmful life habits among children before and during the COVID-19 pandemic. However, confirmations on the changes in physical functions and life habits among children remain inconsistent in the studies on this subject, as it pertains to the COVID-19 pandemic.”
Comment_4
Finally, the whole introduction mentions 2 articles in related to screen time and only 1 article related to sleep but there is nothing discussed about other life habits (quality of life, meals etc.).
Response: Thank you for pointing this out. We agree with the reviewer’s suggestion. We have added this information and cited the relevant literature in the Introduction as requested.
“Regarding dietary changes among children, an increase in the amount of food consumed at home has been reported during the COVID-19 pandemic [9,10]. Moreover, according to a study conducted in Germany, the quality of life among children and adolescents has declined during the COVID-19 pandemic [11].”
Comment_5
Perhaps divide the study design and population to different headings.
The second data point was collected in a 2year period. Were conditions the same throughout the 2 years, or was it different? (Other than citing age as a limitation)
Response: Thank you for pointing this out. Throughout the two years, it was difficult to keep all time and seasons under the same conditions, except for the location of the measurement environment, partly because data measurement had to be discontinued due to repeated declarations of a state of emergency as the number of people infected increased. Thus, we have added the following details to the limitations section.
“Additionally, except in the locations used for the evaluation environments, some participants could not be evaluated under similar conditions, such as time of day and season. Therefore, there may be some residual confounding impacts of age, time of day, and season on the findings.”
Comment_6
How did the authors decide on the effect size values to determine sample size? (and later cite that as a limitation?)
Response: Thank you for pointing this out. According to your suggestion, we have added the following sentences to explain the procedure taken in this effect size values. Additionally, the text listed regarding sample size in the limitations has been removed.
“The effect size of 0.5 was determined based on the mean and standard deviations of the data associated with balance functions from previous studies conducted by Yanovich et al. during the COVID-19 pandemic [32].”
Comment_7
Explain how gait speed is related to physical function.
Response: Thank you for pointing this out. We agree with the reviewer’s suggestion. We have added this information and cited the relevant literature in the Materials and Methods as requested.
“Gait speed is a crucial component for evaluating physical functions, and it has been reported to be a basic indicator of gait development among children, which reflects the status of a healthy individual [24,25]. “
Comment_8
Check footnote under Table 3.
Response: Thank you for pointing this out. This error has been corrected following your comment.
Comment_9
In Line 210 the results of dynamic balance was presented, but the next sentence is about standing on one leg, The sequencing of sentences is confusing. Perhaps follow results with the supporting articles, followed by the discussion.
Response: Thank you for pointing this out. This error has been corrected following your comment.
Comment_10
Line 216, how did the data show limited number of places to engage in physical activity?
Response: Thank you for your valuable suggestion. The expression was a little difficult to convey. In accordance with your comment, we have revised the text for clarity in the Discussion section, as follows.
“As demonstrated in a previous study conducted by Martínez-Córcoles et al. [33], the data obtained in this study showed that the limited time for engaging in social-based physical activities among children and the insufficient opportunities for such individuals to engage in such physical activities during the COVID-19 pandemic may result in decreased balance.”
Comment_11
“Limited opportunities” was mentioned several times as the reason for declining results. Describe limited opportunities in detail.
Response: Thank you for pointing this out. We agree with the reviewer’s suggestion. We have added this information in the Discussion and Conclusions sections, as requested.
“This could be because of the limited opportunities resulting from the decreased engagement in school-based club activities and sports, thereby shortening outdoor playtime, with the main aim being the prevention of the spread of infection by limiting exercise tasks, such as minimizing contact with others when it comes to engaging in physical activities, which results in an irregular life pattern that emerged during the COVID-19 pandemic.”
“These findings suggest that limited opportunities for children to participate in school-based club activities and sports during the COVID-19 pandemic shortened their outdoor playtime periods, with the main aim being the prevention of the spread of COVID-19 infections by limiting exercise tasks, such as minimizing contact with others and limiting their engagement in physical activities.”
Comment_12
The results of Quality of life and meals are presented, but there are no discussions about the reasons there were no changes before and during Covid.
Response: Thank you for pointing this out. We agree with the reviewer’s suggestion. We have added this information in the Discussion section, as requested.
“The quality of life and the number of meals among children, which were assessed using questionnaires based on the Pediatric Quality of Life Inventory (version 4.0) and the number of meals, remain unaffected during the COVID-19 pandemic. This result was similar to those of previous studies comparing children aged between six and seven years before and after the emergency declarations [3]. Therefore, children in Japan may be less sensitive to changes due to the COVID-19 pandemic in terms of quality of life and the number of meals.”
Comment_13
No proper discussion on body fat percentage.
Response: We agree with the reviewer’s comment. This discussion has been corrected in accordance with the reviewer’s comment.
“Moreover, regarding the increase in body fat percentage among children, there was no significant difference in physical activity time and the number of meals, thereby suggesting that irregular lifestyles, such as increased screen time and the insufficient ability to ensure the high quality of exercise among children, may be the cause behind this result.”
Comment_14
There needs to be a stronger justification for including gait analysis in this study (only for gait speed?)
Response: Thank you for pointing this out. We agree with the reviewer’s comment. This Introduction and Materials and Methods has been corrected in accordance with the reviewer’s comment.
“Physical functions refer to the abilities allowing for the performance of physical activities among individuals. Such functions are evaluated using the single leg stance or one-legged stance test as well as the two-step test as measures of balance control, the five times sit-to-stand test (FTSST) as a measure of functional performance, and gait speed as a measure of gait ability [12].”
“Gait speed is a crucial component for evaluating physical functions, and it has been reported to be a basic indicator of gait development among children, which reflects the status of a healthy individual [24,25].”

Reviewer 2 Report
Thank you for the opportunity to read and review the paper.
The paper is very interesting and relevant as it addresses a key health promotion area; Physical functions of children before and during COVID-19.
It is also important that the paper adopted a longitudinal design, considering the vast majority of existing studies used cross-sectional designs.
The introduction is well written, and a reasonable justification was given for the conduct of the study. It established the interlink between the pandemic, associated restrictions and physical functioning.
The study methods were clearly described to a large extent, including the method of measuring parameters such as FTSST, two-step test, single-legged standing time, gait analysis and percentage of body fat, each justified with appropriate literature. The analysis method is also appropriate to establish relationships. Overall, the method is sufficient to allow for replication of the study in a similar context.
It further discusses the major findings and acknowledges its limitations (particularly the small sample size), out of which a recommendation for further study was outlined.
Overall, it is a great study considering the lack of longitudinal studies in the context of the pandemic. However, some minor areas of improvement are recommended below.
Considering there is no mention of a comparison group and the whole study look at relationships/comparisons before & after, I would recommend the general term “longitudinal study” or “prospective longitudinal study” rather than “cohort study”. However, if you think otherwise, you need to justify it as a cohort study under 2.1 (study design and population). Moreover, the main study commenced long before Covid-19 was anticipated (2018), and the current study reports findings as part of the main longitudinal survey.
The fact that Covid-19 is still ongoing, I would recommend replacing it with ‘during Covid-19’ or specifying the month/year, rather than ‘throughout Covid-19”. Otherwise, operationally define what you consider ‘throughout the Covid-19 pandemic’. Additionally, my understanding is that the data during the pandemic was collected once, if not, make it clear how many times the set of data was collected per child throughout the pandemic as stated (June 2020 to June 2022)
It is also worth defining the physical function and life habits if they are referring to different constructs of activity, or just stick to the physical functions.
Please reference the Declaration of Helsinki (page 2 line 80) for readers and the Pediatric Quality of Life Inventory (page 2 line 24) for further information.
Regards
Author Response
Point-by-Point Responses
Reviewer #2
We are grateful to the reviewer for the constructive feedback and for taking the time to evaluate our manuscript. Your comments and suggestions have been immensely helpful in improving our work.
Comment_1
Thank you for the opportunity to read and review the paper.
The paper is very interesting and relevant as it addresses a key health promotion area; Physical functions of children before and during COVID-19.
It is also important that the paper adopted a longitudinal design, considering the vast majority of existing studies used cross-sectional designs.
The introduction is well written, and a reasonable justification was given for the conduct of the study. It established the interlink between the pandemic, associated restrictions and physical functioning.
The study methods were clearly described to a large extent, including the method of measuring parameters such as FTSST, two-step test, single-legged standing time, gait analysis and percentage of body fat, each justified with appropriate literature. The analysis method is also appropriate to establish relationships. Overall, the method is sufficient to allow for replication of the study in a similar context.
It further discusses the major findings and acknowledges its limitations (particularly the small sample size), out of which a recommendation for further study was outlined.
Overall, it is a great study considering the lack of longitudinal studies in the context of the pandemic. However, some minor areas of improvement are recommended below.
Response: Thank you for bringing up these issues. The manuscript has been rechecked carefully by a native English-speaking researcher from a professional editing company, and the necessary changes have been made in accordance with the reviewers’ suggestions. More detailed responses to each of the comments have been prepared and are provided below.
Comment_2
Considering there is no mention of a comparison group and the whole study look at relationships/comparisons before & after, I would recommend the general term “longitudinal study” or “prospective longitudinal study” rather than “cohort study”. However, if you think otherwise, you need to justify it as a cohort study under 2.1 (study design and population). Moreover, the main study commenced long before Covid-19 was anticipated (2018), and the current study reports findings as part of the main longitudinal survey.
Response: We would like to thank the reviewer for the comment. We agree with the reviewer’s suggestion that “longitudinal study” or “prospective longitudinal study” rather than “cohort study”. Therefore, we changed from a “cohort study” to a “prospective longitudinal observational study.”
Comment_3
The fact that Covid-19 is still ongoing, I would recommend replacing it with ‘during Covid-19’ or specifying the month/year, rather than ‘throughout Covid-19”. Otherwise, operationally define what you consider ‘throughout the Covid-19 pandemic’. Additionally, my understanding is that the data during the pandemic was collected once, if not, make it clear how many times the set of data was collected per child throughout the pandemic as stated (June 2020 to June 2022)
Response: We would like to thank the reviewer for pointing this out, and we agree with this comment. As noted by the reviewer, we made the change from "throughout COVID-19" to "during COVID-19". Additionally, we have included the following text to the section describing the data collection process (page 4).
“The data obtained from the participants were evaluated only once before and during the COVID-19 pandemic.”
Comment_4
It is also worth defining the physical function and life habits if they are referring to different constructs of activity, or just stick to the physical functions.
Please reference the Declaration of Helsinki (page 2 line 80) for readers and the Pediatric Quality of Life Inventory (page 2 line 24) for further information.
Response:
Thank you for pointing this out. We also believe that it would be beneficial to provide more data regarding the physical functions and life habits. We have added this information to the Abstract, Introduction, and Materials and Methods sections.
“The compared variables included muscle strength, static and dynamic balance functions, gait speed, body fat percentage, screen and sleep times, quality of life, and physical activity time. During the pandemic, compared to before the pandemic, children had lower levels of dynamic balance functions (p = .039), increased body fat percentages (p < .0001), longer screen time per day (p = .002), and shorter sleep time per day (p < .0001). Between the two periods, there were no significant differences in muscle strength, static balance functions, gait speed, quality of life, and physical activity time.” (Page 1)
“Physical functions refer to the abilities allowing for the performance of physical activities among individuals. Such functions are evaluated using the single leg stance or one-legged stance test as well as the two-step test as measures of balance control, the five times sit-to-stand test (FTSST) as a measure of functional performance, and gait speed as a measure of gait ability [12].” (Page 2)
“This study was conducted in accordance with the Declaration of Helsinki. For the evaluation of the physical functions, lifestyle habits, quality of life questionnaires, and medical examinations, approval was obtained from the Ethics Review Board of the corresponding author’s institution (no. 29002; approval date: October 10, 2017). Written informed consent and assent were obtained from all the children involved in this study as well as from their parents and/or guardians.” (Page 2)

Round 2
Reviewer 1 Report
Thank you to the authors for addressing all the comments, and providing additional details so that explanations are now clearer.
I have two other comments with regards to the manuscript.
1. Kindly clarify, the phrase ‘in this study’ in the sentence below (Line 267), is referring to the current prospective study, OR the study by Martinez-Corcoles et al. Right now, it reads as if it is the former, and thus shows a discrepancy, as the results of the current prospective study, physical activity time was not an influencing factor.
“As demonstrated in a previous study conducted by Martínez-Córcoles et al. [33], the data obtained in this study showed that the limited time for engaging in social-based physical activities among children and the insufficient opportunities for such individuals to engage in such physical activities during the COVID-19 pandemic may result in decreased balance.”
2. Perhaps break down the sentence from Line 345 to 349 as it is too lengthy, and quite confusing.
Author Response
Point-by-Point Responses to Reviewer #1
Comment_1
Thank you to the authors for addressing all the comments, and providing additional details so that explanations are now clearer.
I have two other comments with regards to the manuscript.
Response: Thank you for providing us with your insightful comments regarding our manuscript. We sincerely appreciate the time and effort required to review our study. Your comments and suggestions have helped improve our work immensely.
Changes have been made in accordance with the reviewers’ suggestions, and the manuscript has also been carefully rechecked by a native English-speaking researcher from a professional editing company. Detailed responses to each of the comments have been prepared and are provided below.
Comment_2
Kindly clarify, the phrase ‘in this study’ in the sentence below (Line 267), is referring to the current prospective study, OR the study by Martinez-Corcoles et al. Right now, it reads as if it is the former, and thus shows a discrepancy, as the results of the current prospective study, physical activity time was not an influencing factor.
“As demonstrated in a previous study conducted by Martínez-Córcoles et al. [33], the data obtained in this study showed that the limited time for engaging in social-based physical activities among children and the insufficient opportunities for such individuals to engage in such physical activities during the COVID-19 pandemic may result in decreased balance.”
Response: Thank you for pointing this out. We agree with the reviewer’s suggestion, and the text has been corrected accordingly.
“A previous study conducted by Martínez-Córcoles et al. [33] found that limited time and opportunities for children to engage in social-based physical activities may result in decreased balance among them.”
Comment_3
Perhaps break down the sentence from Line 345 to 349 as it is too lengthy, and quite confusing.
Response: We agree with the reviewer’s comment. The sentence has been split and revised for clarity.
“During the COVID-19 pandemic, children’s exercise tasks were often limited in order to prevent the spread of infection. Specifically, children had limited engagement in outdoor physical activities, school-based club activities, and sports. This led to inadequate opportunities for physical education, which significantly affected the physical functions among children in Japan.”
